# Immunotherapy in Melanoma: Recent Advances and Future Directions

**DOI:** 10.3390/cancers15041106

**Published:** 2023-02-09

**Authors:** Andrew Knight, Lilit Karapetyan, John M. Kirkwood

**Affiliations:** 1Department of Medicine, Division of General Internal Medicine, University of Pittsburgh Medical Center, Pittsburgh, PA 15213, USA; 2Department of Cutaneous Oncology, H. Lee Moffitt Cancer Center and Research Institute, Tampa, FL 33612, USA; 3Department of Medicine, Division of Hematology/Oncology, University of Pittsburgh Medical Center, Hillman Cancer Center, Pittsburgh, PA 15213, USA; 4Tumor Microenvironment Center, UPMC Hillman Cancer Center, Pittsburgh, PA 15232, USA

**Keywords:** melanoma, immunotherapy, immune checkpoint blockade, LAG-3, TLR-9, STING, T-VEC, fecal microbiota transplant, adoptive cell therapy

## Abstract

**Simple Summary:**

Immunotherapy has demonstrated the ability to reduce the risk of recurrence for melanoma following surgical resection and improve survival in patients with unresectable disease. Despite the significant advances made in the treatment of patients with melanoma, many patients will have recurrence or progression of disease despite currently available treatments. It is, therefore, critical to identify additional immunotherapy agents that will provide clinical benefit on their own or in combination with existing therapies. In this review, we explore the recent history of immunotherapy development, highlight landmark trials, and discuss promising treatments for patients refractory to current therapies.

**Abstract:**

The use of immunotherapy in the treatment of advanced and high-risk melanoma has led to a striking improvement in outcomes. Although the incidence of melanoma has continued to rise, median survival has improved from approximately 6 months to nearly 6 years for patients with advanced inoperable stage IV disease. Recent understanding of the tumor microenvironment and its interplay with the immune system has led to the explosive development of novel immunotherapy treatments. Since the approval of the therapeutic cytokines interleukin-2 and interferon alfa-2 in the 1990s, the development of novel immune checkpoint inhibitors (ICIs), oncolytic virus therapy, and modulators of the tumor microenvironment have given way to a new era in melanoma treatment. Monoclonal antibodies directed at programmed cell death protein 1 receptor (PD-1) and its ligand (PDL-1), cytotoxic T-lymphocyte-associated protein 4 (CTLA-4), and lymphocyte-activation gene 3 (LAG-3) have provided robust activation of the adaptive immune system, restoring immune surveillance leading to host tumor recognition and destruction. Multiple other immunomodulatory therapeutics are under investigation to overcome resistance to ICI therapy, including the toll-like receptor-9 (TLR-9) and 7/8 (TLR-7/8) agonists, stimulator of interferon genes (STING) agonists, and fecal microbiota transplantation. In this review, we focus on the recent advances in immunotherapy for the treatment of melanoma and provide an update on novel therapies currently under investigation.

## 1. Introduction

Prior to the introduction of immunotherapy for the treatment of advanced melanoma, outcomes were generally poor despite the application of many cytotoxic agents and combinations. The median survival of unresectable metastatic disease was 6–9 months [1,2,3]. Following the approval of first- and second-generation checkpoint blockade inhibitor immunotherapy agents, survival has significantly improved among these patients, with long-term follow-up analysis demonstrating a median survival of nearly 6 years for the combination of CTLA4 and PD1 blockade [4]. In 1995, interferon-α2b was the first agent approved for adjuvant therapy by the FDA after demonstrating its ability to reduce disease recurrence and death in high-risk patients. In 1998, interleukin-2 (IL-2) was the first immunotherapy agent approved for use in patients with advanced metastatic melanoma. However, the use of IL-2 was limited by significant toxicity. As such, its use was largely restricted to a limited number of experienced specialty centers [5]. In 2011, the first CTLA-4 immune checkpoint inhibitor, ipilimumab, was approved for treatment of advanced inoperable metastatic melanoma. Since then, this agent was approved for adjuvant therapy of high-risk node-metastatic resected melanoma, and anti-programmed cell death protein 1 (PD-1) monoclonal antibodies, pembrolizumab and nivolumab, have been approved in metastatic and adjuvant settings. In March of 2022, a combination of the anti-LAG-3 monoclonal antibody, relatlimab, in a fixed-dose combination with nivolumab was approved for the treatment of metastatic melanoma. A timeline of these approvals can be seen in Figure 1. Despite these recent advances, many patients currently treated with immunotherapy experience progression of disease indicating an ongoing need for a novel therapy for advanced melanoma. In this review, we highlight the recent advances in immunotherapy for the treatment of melanoma in addition to future potential therapies.

## 2. Anti-CTLA-4

Cytotoxic T lymphocyte-associated protein 4 (CTLA-4) plays a crucial role in immune tolerance and has been strongly implicated in anti-tumor immune responses [6,7,8]. CTLA-4 is a cell surface receptor and CD28 homolog expressed on both regulatory and conventional T cells that is upregulated following T cell activation [9]. The binding of B7 to CTLA-4 initiates an inhibitory signaling cascade resulting in the suppression of T cell function [10]. Anti-CTLA-4 monoclonal antibodies prevent CD28 binding to B7, resulting in increased T cell activation and enhanced immune recognition of cancer cells [7].

Ipilimumab is a monoclonal antibody that inhibits the function of anti-cytotoxic T lymphocyte-associated protein 4 (CTLA-4). Ipilimumab first demonstrated clinical efficacy in melanoma during a trial in which patients with metastatic melanoma were treated with ipilimumab (3 mg/kg) with or without a gp100 vaccine compared to the gp100 vaccine alone. Patients who received ipilimumab demonstrated improved overall survival compared to patients who received only gp100 vaccine, with a median overall survival of 10.0 for ipilimumab plus gp100, 10.1 for ipilimumab alone, and 6.4 months for gp100 alone. Grade ≥ 3 adverse events occurred in 10–15% of patients, with the most common toxicities reported being diarrhea, dermatitis, and fatigue [11]. Based on the results of this trial, ipilimumab (3 mg/kg) received FDA approval for the treatment of unresectable stage IV melanoma in 2011.

Following these results, ipilimumab in combination with dacarbazine was compared to dacarbazine alone in patients with previously untreated metastatic melanoma. A 10 mg/kg dose of ipilimumab was utilized in this trial due to evidence of a dose-dependent response rate in a preceding phase 2 trial (objective response rate 11.1% for 10 mg/kg vs. 4.2% for 3 mg/kg). Ipilimumab plus dacarbazine demonstrated a significantly longer overall survival when compared to dacarbazine alone (11.2 vs. 9.1 months). However, the grade ≥ 3 adverse event rate was significantly higher in the group of patients that received ipilimumab with dacarbazine compared to dacarbazine alone (56.3% vs. 27.5%) [12].

Given the prior results indicating the clinical efficacy of ipilimumab at 3 mg/kg and 10 mg/kg, the doses were compared in a phase 3 trial in patients with metastatic or unresectable melanoma. A dose of 10 mg/kg demonstrated improved overall survival (median overall survival 15.7 months vs. 11.5 months) and recurrence-free survival for 10 mg/kg compared to 3 mg/kg. However, the incidence of treatment-related adverse events was significantly higher in the 10 mg/kg group with grade ≥ 3 adverse events occurring in 34.0% compared to 18.2% in the 3 mg/kg group [13].

After demonstrating clinical efficacy in the metastatic setting, ipilimumab was investigated as adjuvant therapy in several trials. For fully resected stage III melanoma, ipilimumab 10 mg/kg improved overall survival compared to placebo (hazard ratio 0.72, confidence interval 0.58 to 0.88) [14]. Due to the high incidence of serious adverse events seen in patients treated with ipilimumab at the dose of 10 mg/kg, ipilimumab 3 and 10 mg/kg were compared to high-dose interferon alfa-2b (HDI) for resected high-risk melanoma. Ipilimumab 3 mg/kg demonstrated improved overall survival compared to HDI (hazard ratio 0.78, confidence interval 0.61 to 0.99). Trends in favor of improved survival with ipilimumab 10 mg/kg compared to HDI did not reach statistical significance. Additionally, significantly more toxicity was encountered in patients treated with ipilimumab 10 mg/kg compared to 3 mg/kg (46.3% vs. 28.5%) [15]. Ipilimumab received FDA approval as adjuvant therapy for fully resected stage III melanoma in 2015 at the higher dosage of 10 mg/kg, but the more recent data from E1609 argue for consideration of 3 mg/kg over the original dosage.

The anti-CTLA4 antibody tremelimumab was also evaluated as a potential treatment for advanced melanoma. However, in an open-label trial, tremelimumab failed to demonstrate a survival benefit when compared to standard-of-care dacarbazine or temozolomide. The availability of ipilimumab may have confounded this, and tremelimumab is not currently used in clinical practice for the treatment of melanoma [16].

## 3. Anti-PD-1

The programmed cell death protein 1 (PD-1) receptor is expressed on the surface of lymphocytes, while its ligand (PD-L1) is widely expressed on both immune and non-immune cells [17]. The binding of PD-1 to PD-L1 results in decreased T cell activity, promoting self-tolerance [18]. PD-L1 is overexpressed on a wide variety of tumors, including melanoma, resulting in immune evasion by cancer cells. Inhibition of PD-1 restores standard immune recognition of malignant cells and subsequent immune-mediated destruction [17].

### 3.1. Pembrolizumab

The anti-tumor effect of PD-1 inhibition has been demonstrated using two monoclonal antibodies specific for PD-1, pembrolizumab and nivolumab. KEYNOTE-001, a phase 1 trial investigating the use of pembrolizumab in multiple advanced solid tumors, first signaled potential clinical efficacy for the treatment of advanced melanoma [19]. These results were further investigated in KEYNOTE-002, which demonstrated the efficacy of pembrolizumab after progression on ipilimumab compared to standard-of-care chemotherapy. Pembrolizumab resulted in improved progression-free survival at both 2 mg/kg and 10 mg/kg (hazard ratio 0.57 and 0.50, respectively), with 6-month progression-free survival of 34% in the pembrolizumab 2 mg/kg group, 38% in the 10 mg/kg group, and 16% in the chemotherapy group. Additionally, fewer grade ≥ 3 adverse events were reported for patients treated with pembrolizumab compared to chemotherapy (11% for pembrolizumab 2 mg/kg, 14% for pembrolizumab 10 mg/kg, and 26% for chemotherapy) [20]. The results of this trial established pembrolizumab as a new standard of care.

Pembrolizumab was subsequently compared to ipilimumab for the treatment of advanced melanoma in the KEYNOTE-006 trial and demonstrated improved progression-free survival (hazard ratio 0.58) and overall survival (hazard ratio 0.63). Pembrolizumab was also associated with less toxicity, with a grade ≥ 3 adverse event rate of 13.3% compared to 19.9% [21]. Following these results, pembrolizumab was granted FDA approval for the treatment of unresectable or metastatic melanoma in the front-line setting.

Pembrolizumab has demonstrated efficacy in the adjuvant setting for high-risk stage III melanoma. KEYNOTE-054 compared pembrolizumab to placebo for postoperative treatment of high-risk resected stage III melanoma. High-risk stage III disease was defined as patients with stage IIIA disease with >1mm nodal metastasis or stage IIIB-C. Results demonstrated significantly longer recurrence-free survival than placebo (75.4% vs. 61.0% 1-year recurrence-free survival, respectively). Adverse events occurred at a similar rate, as was seen with the adjuvant use of nivolumab in the Checkmate 238 trial with grade ≥ 3 adverse event rate of 14.7% [22,23].

Following results demonstrating its clinical benefit as adjuvant therapy for patients with stage III melanoma, pembrolizumab was investigated for use as adjuvant therapy in patients with high-risk stage II disease compared to placebo under KEYNOTE-716—A double-blind, randomized, phase 3 study. High-risk stage II melanoma was defined as stage IIB or IIC melanoma. Results demonstrated a significant reduction in disease recurrence or death (hazard ratio 0.61, confidence interval 0.45–0.82) with toxicity similar to that seen in prior trials [24]. Following these results, pembrolizumab received FDA approval in 2021 for adjuvant treatment of stage IIB or IIC melanoma following complete resection.

There is also interest in neoadjuvant immunotherapy as an alternative strategy for resectable disease. Neoadjuvant immunotherapy offers several potential advantages, including antigen exposure during immunotherapy treatment before resection, activation of in situ tumor-infiltrating lymphocytes, and potentially decreased surgical morbidity due to preoperative reduction of tumor size. Results from the recent SWOG S1801, a randomized phase II study, compared neoadjuvant pembrolizumab with adjuvant pembrolizumab for patients with Stage IIIB-IV resectable melanoma. Of the 313 patients enrolled, 154 were randomized to receive neoadjuvant pembrolizumab every 3 weeks for 3 cycles, followed by surgical resection, then 15 additional cycles of postoperative pembrolizumab. The control group of 159 patients was randomized to receive standard-of-care adjuvant pembrolizumab with surgery upfront followed by 18 cycles of pembrolizumab every 3 weeks. Results indicated a significant benefit of pembrolizumab upon event-free survival (EFS, recurrence, or progression) administered in the neoadjuvant setting (hazard ratio 0.59, confidence interval 0.40–0.86) over standard postoperative ICI therapy that was essentially identical for the two arms—with one administered pre-surgically for three cycles and the other administered entirely postoperatively [25].

One promising strategy to improve response rates and outcomes with pembrolizumab in the neoadjuvant setting involves utilizing the anti-angiogenic/multiple RTK inhibitor lenvatinib. Results of the single-arm phase II trial NeoPeLe have indicated the potential for neoadjuvant pembrolizumab in combination with lenvatinib. A total of 20 patients with resectable stage III disease were treated with 6 weeks of pembrolizumab and lenvatinib followed by lymph node dissection. Results showed at least a partial pathologic response in 75% (15/20) of patients, with a complete pathologic response observed in 40% (8/20). One-year event-free survival, defined as recurrence or progression, was 80% [26]. The results of these trials provide clinical evidence supporting the potential role of neoadjuvant immunotherapy in resectable melanoma.

### 3.2. Nivolumab

Nivolumab is a fully human IgG4 monoclonal antibody directed against PD-1. In 2012, results of a phase 1 trial were published, demonstrating the activity of BMS-936558, later named Nivolumab, in patients with advanced solid tumors, including melanoma. Following these results, nivolumab received accelerated FDA approval in 2014 for the treatment of advanced melanoma [27,28].

These results were expanded upon in the CheckMate 037 trial, which compared nivolumab to chemotherapy (investigator’s choice of dacarbazine or carboplatin and paclitaxel) for advanced melanoma. Results demonstrated an improved objective response rate (27% vs. 10%) and durable responses (32 vs. 13 months). No statistically significant differences were observed in progression-free survival or overall survival. Nivolumab was better tolerated than chemotherapy, with a grade ≥ 3 adverse event rate of 14% for nivolumab compared to 34% for chemotherapy [29].

Nivolumab was then compared to standard-of-care dacarbazine in a previously untreated population of patients with advanced melanoma in CheckMate 066. In this population, nivolumab demonstrated significant improvement in overall survival (1 year overall survival 72.9% vs. 42.1%; hazard ratio for death 0.42) and progression-free survival (5.1 months vs. 2.2 months; hazard ratio for progression or death 0.43) [30]. A 5-year follow-up analysis of these patients has further supported the improvement in long-term outcomes of progression-free survival, overall survival, and objective response rate for patients treated with nivolumab compared to dacarbazine [31].

Nivolumab has also been approved for use as adjuvant therapy. CheckMate 238 compared ipilimumab to nivolumab after complete resection for patients with stage IIIB, IIIC, or IV melanoma. Treatment with nivolumab demonstrated improved recurrence-free survival compared to ipilimumab (hazard ratio 0.66 at 2 years) [32]. In contrast, ipilimumab compared to placebo in the adjuvant setting demonstrated a hazard ratio of 0.75 for recurrence or death [33]. Taken together, these results suggest that nivolumab, compared to placebo, would have an overall benefit in terms of relapse that is associated with a hazard ratio of approximately 0.5 for recurrence or death as adjuvant therapy. Nivolumab has also shown a favorable side effect profile compared to ipilimumab, with a grade ≥ 3 adverse event rate of 14.4% with nivolumab compared to 45.9% for ipilimumab [32,34].

## 4. Ipilimumab and Nivolumab

Ipilimumab and nivolumab, in combination, have demonstrated synergistic effects on the immune response against advanced melanoma beyond that seen with a single agent. A phase I study of ipilimumab in combination with nivolumab utilizing a standard 3 + 3 dose-escalation design determined nivolumab dosed at 1 mg/kg and ipilimumab dosed at 3 mg/kg to be the maximum doses that were associated with an acceptable level of adverse events [35]. Following these results, a combination of ipilimumab and nivolumab was compared to ipilimumab monotherapy in a phase II trial for treatment-naïve patients with unresectable stage III or IV disease. For patients treated with a combination of nivolumab and ipilimumab, treatment was initiated with both nivolumab (1 mg/kg) and ipilimumab (3 mg/kg) once every 3 weeks for four doses. Ipilimumab was then discontinued, and patients continued nivolumab monotherapy (3 mg/kg) until disease progression or death. For patients in the ipilimumab monotherapy group, ipilimumab (3 mg/kg) was infused every 3 weeks. Patients who received combination therapy had an improved objective response rate compared to ipilimumab monotherapy (61% vs. 11%, respectively) with complete response rates of 22% and 0%, respectively. Combination therapy was also associated with improved progression-free survival with a hazard ratio of progression or death of 0.40 (CI 0.23–0.68) [36]. Following these results, the original combination of ipilimumab and nivolumab received FDA approval for advanced melanoma.

CheckMate-067 phase III trial investigated the clinical efficacy of a combination of ipilimumab and nivolumab at the forgoing dosages compared to each agent alone. Patients with unresectable stage III or IV melanoma were randomized 1:1:1 to nivolumab plus ipilimumab, nivolumab alone, or ipilimumab alone. A combination of nivolumab and ipilimumab was administered using the same dosing strategy utilized during the phase II trial. Findings demonstrated that combination therapy improved progression-free survival and overall survival over ipilimumab compared to nivolumab monotherapy [37]. Long-term outcomes at 6.5 years have shown a median overall survival of 72.1 months for the ipilimumab and nivolumab group, 36.9 months for nivolumab monotherapy, and 19.9 months with ipilimumab monotherapy. Treatment-related adverse events that led to discontinuation of the study drug occurred in 7.7% of the patients in the nivolumab group compared to 36.4% of those in the combination therapy group. Grade ≥ 3 adverse events occurred in 59% of the combination ipilimumab and nivolumab group, 24% with nivolumab monotherapy, and 28% with ipilimumab monotherapy [4].

For patients with advanced melanoma with targetable BRAF mutations, treatment with BRAF/MEK inhibitors and ICIs are both potential front-line treatment options. However, retrospective analyses have indicated a survival benefit for patients who initiated therapy with ICI first [38]. The prospective randomized controlled phase III trial, DREAMseq, was conducted to determine which treatment sequence has the best efficacy. In the trial, therapy was initiated with either combination nivolumab/ipilimumab or dabrafenib/trametinib. At disease progression, the alternate therapy was administered. Of the 265 patients enrolled, 73 proceeded to the second phase of the trial. The study was stopped early by the DSMB because a significant overall survival benefit was seen in patients treated with the combination of nivolumab/ipilimumab first over dabrafenib/trametinib first (2-year overall survival 71.8% vs. 51.5%, respectively). The two treatments’ objective response rates were similar when used in the front-line setting (46 vs. 43%). When used in the second line setting, dabrafenib/trametinib demonstrated a similar objective response rate of 47.8%. However, the combination nivolumab/ipilimumab revealed a significantly lower objective response rate of 29.6% after progression on initial therapy with dabrafenib/trametinib [39]. Further adding to the evidence for a benefit of combination immunotherapy followed by targeted therapy, a smaller phase II trial known as SECOMBIT demonstrated similar results when comparing the sequential treatment with encorafenib/binimetinib. Administered in the front-line setting, nivolumab/ipilimumab had an objective response rate of 44.9%, but when utilized after progression on encorafenib/binimetinib, the objective response rate for nivolumab/ipilimumab was only 25.7% [40].

Although combination immune checkpoint blockade has significantly improved survival for patients with advanced melanoma, approximately half of all patients treated suffer grade ≥ 3 adverse events. As a result, an investigation is underway to determine if the toxicity of combination ICI treatment can be reduced while preserving its efficacy. One such approach involves utilizing tocilizumab, a humanized IL-6 receptor-blocking antibody, with ipilimumab and nivolumab. High levels of IL-6 have been associated with poor prognosis and an increased likelihood of developing immune-related adverse events during ICI treatment in patients with melanoma [41,42,43]. A phase II trial in patients with advanced melanoma studied the combination of tocilizumab (4 mg/kg), ipilimumab (1 mg/kg), and nivolumab (3 mg/kg). Results indicate an objective response rate of 70% at the 6-month follow-up and a grade 3/4 immune-related adverse event rate of only 25% [44]. While the results are promising and suggest that the addition of tocilizumab may result in reduced toxicity, confirmation in a larger randomized controlled study is needed.

Ipilimumab and nivolumab in combination have demonstrated robust clinical activity for advanced melanoma and have been investigated for potential use as adjuvant therapy. The IMMUNED trial was the first double-blind, randomized, placebo-controlled phase II trial to study the impact of a combination of ipilimumab and nivolumab compared to nivolumab monotherapy or placebo in patients with resected stage IV melanoma. Results demonstrated a significant benefit in recurrence-free survival of combination ipilimumab and nivolumab compared to placebo with a 1-year recurrence-free survival of 75% and 32%, respectively. However, the combination therapy of ipilimumab and nivolumab did not show a statistically significant benefit compared to nivolumab monotherapy, with 1-year recurrence-free survival rates of 75% (CI 61.0–84.9%) and 52% (CI 38.1–63.9%), respectively [45].

The phase III trial CheckMate 915 further explored the role of combination immunotherapy in the adjuvant setting. Ipilimumab and nivolumab were compared to nivolumab monotherapy in patients with resected stage IIIB-IV melanoma. A total of 1844 patients were randomized 1:1 to receive ipilimumab and nivolumab or nivolumab monotherapy for one year post-resection. In contrast to the dosing utilized for the combination in the metastatic setting, the dosage of ipilimumab was reduced (1 mg/kg) and the frequency elongated to 6 weeks, with the duration of the combination treatment extended to the entire one-year treatment period. Similar to the results of the IMMUNED trial, the addition of ipilimumab to nivolumab for adjuvant treatment did not demonstrate improved recurrence-free survival compared to nivolumab monotherapy (2-year recurrence-free survival 64.6% and 63.2%, respectively) [46].

## 5. Nivolumab and Relatimab

Following the clinical success of nivolumab utilized in combination with ipilimumab, other combinations of immune checkpoint inhibitors have been explored. One such combination has utilized an inhibitor of the lymphocyte-activation gene 3 (LAG-3) in addition to nivolumab. LAG-3 is a CD4 homolog expressed on activated CD4+ and CD8+ T lymphocytes that binds to MHC II with significantly higher affinity than CD4 [47]. Binding of LAG-3 to MHC-II results in negative regulation of T cell proliferation and function [48,49]. As melanoma cells often express MHC class II, binding of LAG-3 to tumor-infiltrating T cells can result in potent immune suppression and, ultimately, immune evasion by melanoma [50]. LAG-3 positive T cells are present amongst tumor-infiltrating leukocytes within melanoma, leading to the hypothesis that inhibition of LAG-3/MHC II binding will result in improved immune response, T cell proliferation, and ultimately immune destruction of melanoma cells [51].

Relatimab, formerly known as BMS-986016, is a first-in-class human IgG4 LAG-3 blocking antibody. In 2016, the first clinical results were released from a phase 1/2a study utilizing the combination of relatimab and nivolumab in patients with prior progression on anti-PD-1 therapy, with or without anti-CTLA-4 therapy. Of the 61 patients evaluated, objective response rate was 11.5% (one complete, six partial) with a grade ≥ 3 adverse event rate of 4.4%. The low objective response rate was attributed partly to the cohort’s heavily pre-treated nature. However, the demonstration of limited efficacy in a population that had progressed on prior PD-1 therapy led to further focus upon evaluation in a first-line phase 2/3 trial [52,53].

RELATIVITY-047 was a phase 2/3 trial that evaluated the combination of relatimab and nivolumab compared to standard-of-care nivolumab in patients with untreated unresectable stage III or IV disease. The primary outcome measure was progression-free survival. Results demonstrated that the combination therapy improved progression-free survival with a median progression-free survival of 10.1 months with relatimab and nivolumab compared to 4.6 months with nivolumab monotherapy. Grade ≥ 3 adverse events occurred in 18.9% of the relatimab–nivolumab group compared with 9.7% of patients in the nivolumab group. Subgroup analysis included LAG-3 expression (>1% or <1%) and PD-L1 expression (>1% or <1%). Progression-free survival was significantly longer for patients with PD-L1 positive tumors and those with LAG-3 positive tumors. Subgroup analysis also revealed that patients with characteristics associated with worse prognosis, including visceral metastases, high tumor burden, elevated levels of serum LDH, or mucosal melanoma had a significant benefit from the relatimab–nivolumab combination therapy compared to nivolumab alone [54]. Following these results, relatimab used in fixed-dose combination with nivolumab received FDA approval for metastatic or unresectable melanoma.

Nivolumab and relatimab have also been studied in combination as neoadjuvant and adjuvant therapy for patients with fully resectable stage III or IV disease. During a phase II trial, 29 patients were treated with nivolumab and relatimab for two 4-week cycles before surgery and ten additional cycles following resection. Results indicated a pathologic complete response rate (pCR) of 59% and near pCR (<10% of viable tumor) of 7%. For patients with a major pathologic response (pCR or near pCR), the 1- and 2-year recurrence-free survival rates were 100% and 92%, respectively. For patients without a major pathologic response, the 1- and 2-year recurrence-free survival rates were 88% and 55%, respectively. Treatment was associated with a grade ≥ 3 adverse event rate of 26%, all of which occurred during the adjuvant phase of therapy [55]. These promising results stand in contrast to the lack of benefit seen with the addition of ipilimumab to adjuvant nivolumab.

## 6. TLR-9 Agonists

Toll-like receptors (TLRs) are a family of receptors predominately expressed on immune cells that recognize conserved molecular motifs common to pathogenic organisms known as pathogen-associated molecular patterns (PAMPS) [56]. Activation of TLRs results in the activation of innate immunity and subsequently elicits an adaptive immune response. TLR-9 is an intracellular receptor present on the surface of the endoplasmic reticulum and intracellular vesicles that is translocated to endosomes allowing exposure to PAMPs [57]. TLR-9 recognizes nucleic acid PAMPs, specifically unmethylated CpG oligodinucleotides, based on sequence and secondary structure [58,59]. Activation of TLR-9 triggers a downstream intracellular signaling cascade that results in NFkB signaling that releases a type I IFN secretion in the tumor microenvironment [60,61]. Type I interferon then results in the activation of CD8+ T cells and improved antigen presentation, stimulating an anti-tumor response (Figure 2). Pre-clinical data suggest that the activation of TLR-9 results in the recruitment of T cells to the tumor microenvironment and can convert non-inflamed tumors to inflamed tumors [62].

Given the potential of TLR-9 activation to trigger an anti-tumor immune response, TLR-9 agonists have been trialed as therapeutics for a wide range of malignancies, including melanoma, basal cell carcinoma, lymphoma, and renal cell carcinoma [63,64,65,66]. TLR-9 agonists are typically administered intra-lesionally or systemically via subcutaneous injection. TLR-9 agonists have also been studied in combination with chemotherapy, radiation, or immunotherapies [67]. The degree of inflammation within the tumor microenvironment (TME) assessed pathologically as tumor-infiltrating lymphocytes (TILs) has been positively correlated with response to ICIs [68,69]. Therefore, due to their ability to increase TIL response, TLR-9 agonists are an attractive treatment for patients with ICI refractory disease [67].

Although no TLR-9 agonists currently have received FDA approval for the treatment of melanoma, several clinical trials have demonstrated potential clinical benefit and are undergoing further investigation. Tilsotolimod (IMO 2125) is a synthetic TLR-9 agonist studied in intratumor injection in combination with systemic ipilimumab in advanced melanoma refractory to PD-1 inhibitors. Of the 18 patients that received treatment, 9 were treated at the recommended phase II dose of which 6 experienced clinical benefit. Additionally, biopsies of uninjected tumors from responding patients showed expression of CD56+ and Ki67+ effector CD8+T cells indicative of a remote effect. Of the 18 patients treated, 4 developed immune-related adverse events [70].

SD-101 is a TLR-9 agonist that has been studied in combination with pembrolizumab in patients with advanced melanoma without prior exposure to anti-PD-1 therapy under a phase IB/II study (SYNERGY-001/KEYNOTE-184). In total, 86 patients were treated with pembrolizumab in combination with an intratumor injection of SD-101 at either 2 mg or 8mg per lesion with an objective response rate of 71% and 49%, respectively. Immune-related adverse events occurred in 19% of patients with the most common adverse events related to SD-101 injection being headache, fatigue, malaise, and myalgias [71].

Vidutolimod (CMP-001) is a TLR-9 agonist packaged within a virus-like particle that has been studied in combination with pembrolizumab as neoadjuvant therapy for fully resected stage III melanoma and advanced melanoma. A phase II neoadjuvant trial studied intratumoral CMP-001 in combination with pembrolizumab among 30 patients with stage IIIB-D melanoma prior to resection. Pathological responses were seen in 70% (21/30) of the patients that received treatment. Grade ≥ 3 immune adverse events occurred in three patients leading to discontinuation of CMP-001 [72].

CMP-001 was also studied in advanced melanoma under a two-part phase Ib study of patients with metastatic or unresectable disease that had previously progressed on anti-PD-1 therapy. CMP-001 was studied as monotherapy (N = 40) and in combination with pembrolizumab (N = 159). The best objective response rate for combination therapy was 23.5% (23/98) compared with 17.5% (7/40) for CMP-001 monotherapy. Combination therapy was also associated with a more durable response than CMP-001 monotherapy [73]. These results signal the potential utility for application of TLR-9 agonists in patients with PD-1 refractory disease.

## 7. T-VEC

The cutaneous location of melanoma primary lesions and the propensity of melanoma to develop regional cutaneous in-transit metastases opens the opportunity for intra-lesional therapy with minimal procedural risk compared to inoculation of visceral solid tumors in other malignancies. One such intra-lesional therapy is talimogene laherparepvec (T-VEC, previously known as OncoVECGM-CSF). T-VEC is a genetically modified oncolytic HSV-1 virus. T-VEC was developed by deletion of the ICP34.5 gene encoding for the herpes neurovirulence factor, resulting in a loss of the ability to replicate in neurons while preserving the ability to infect and replicate within tumor cells. Additionally, the virus has been modified by inserting tumor-specific promotors resulting in tumor-specificity [74]. The virus was further modified to include the gene encoding for human GM-CSF. GM-CSF is released upon lysis of tumor cells, resulting in an improved immune response via recruitment of APC to the tumor microenvironment and enhancement of dendritic cell function. These APCs are then exposed to tumor antigens and stimulate a T cell response against the tumor [75]. Intratumoral injection of GM-CSF demonstrated limited clinical efficacy as monotherapy [76]. However, GM-CSF encoding strains of HSV-1 have been shown to produce a more robust immune response than HSV-1 strains without GM-CSF expression in pre-clinical models [77]. In addition to a local effect on the tumor, there is evidence that intra-tumor injection can result in an abscopal effect and distant immune response at other disease sites [76,78].

A phase II study was conducted in a heavily pre-treated (74%) population of patients with unresectable stage IIIc-IV disease. Of the 50 patients that participated, 13 (26%) had an objective response, with 8 of 13 patients experiencing complete responses. Following these results, a phase III trial was conducted comparing T-VEC to GM-CSF (administered intralesionally). Of the 436 patients with unresectable stage IIIb-IV disease, T-VEC was associated with a significantly higher objective response rate than GM-CSF (26.4% for T-VEC compared to 5.7% for GM-CSF). Durable response rates were also higher for the patients treated with T-VEC (16.3%) than GM-CSF (2.1%). No statistically significant difference in overall survival was observed. A 5-year follow-up demonstrated similar results. T-VEC is generally well tolerated with a grade ≥ 3 adverse event rate of 11% with fatigue, pyrexia, and chills being the most common adverse events [76,79]. Following these results, T-VEC received FDA approval for unresectable melanoma in 2015. T-VEC remains the only FDA-approved oncolytic viral therapy to date.

The advent of novel immune checkpoint inhibitors has increased interest in utilizing ICI in combination with T-VEC to determine if the immune response can be potentiated. Several clinical trials have investigated the combination of anti-PD-1 antibodies in combination with T-VEC. Early clinical data suggested the therapies may have synergistic effects [80,81]. The MASTERKEY-265 phase 1b studied the combination of T-VEC and pembrolizumab for unresectable stage IIIB-IV melanoma. The combination treatment was associated with a grade ≥ 3 adverse event rate of 33% (7/21 patients) and a confirmed objective response rate of 48% [80]. Subsequently, a phase III trial was conducted in patients with stage IIIB-IVM1c comparing pembrolizumab in combination with T-VEC vs. pembrolizumab plus placebo injection. A total of 692 patients with injectable lesions were recruited. Median progression-free survival for combination treatment was 14.3 months for the combination treatment arm compared with 8.5 months for pembrolizumab plus placebo (hazard ratio for progression or death 0.86, 95% confidence interval 0.71, 1.04, *p* = 0.13). Median overall survival was not reached for the T+P arm and was 49.2 months for the pembrolizumab plus placebo arm (hazard ratio for death 0.96, 95% confidence interval 0.76, 1.24, *p* = 0.74). Overall, the differences in progression-free survival and overall survival were not statistically significant. Safety was comparable between the two groups with a grade ≥ 3 adverse event rate for patients treated with combination therapy vs. pembrolizumab plus placebo of 46.7% and 44.0%, respectively.

## 8. Adoptive Cell Therapy

Adoptive cell therapy (ACT) utilizing tumor-infiltrating lymphocytes (TILs) is a cellular immunotherapy that involves extracting lymphocytes from tumor tissue, expanding this population, and infusing them back into the patient, where they can recognize and attack tumor cells. To isolate TILs, a tumor is surgically resected, then dissected into fragments, and incubated in media containing IL-2 resulting in TIL outgrowth and expansion. The TILs can then be tested for tumor reactivity by incubating the TILs together with autologous tumor cells, after which the TIL populations with the highest cytokine release are selected for rapid expansion. These populations are then incubated with an anti-CD3 antibody, IL-2, and irradiated allogeneic feeder cells. The time from tumor resection to TIL infusion takes approximately 6 weeks [82]. Alternatively, the TIL can be expanded directly after isolating tumor samples without selection for populations based on tumor reactivity. This approach is referred to as the “young TIL” protocol and has the advantage of a higher success rate and faster time to generate a clinically infusible product [83,84]. Clinical success rates between the two approaches are similar [85,86,87]. Following the production of a TIL product, patients are treated with non-myeloablative lymphodepleting chemotherapy with cyclophosphamide and fludarabine. After lymphodepletion, the TIL product is infused into the patients followed by systemic IL-2 (Figure 2).

TIL treatment was first studied clinically as therapy for melanoma over 20 years ago in combination with high-dose IL-2, before modern checkpoint blockade immunotherapy was introduced. Objective response rates of up to 72% were observed, with 10–20% of patients achieving complete responses [83,84,88,89]. Since then, there has been considerable investigation into optimizing the ACT process. The lymphodepletion chemotherapy regimen used in the original ACT trials has been studied in combination with total body irradiation with evidence of increased objective response rate but with increased toxicity [90]. The optimal dose of IL-2 after TIL infusion needs to be better established with amounts ranging from 100,000 to 720,000 IU/kg for 6 to 15 doses administered [84]. Additionally, the role of a combination of ACT and ICI therapy is also under investigation. ACT has been utilized in combination with ipilimumab in 13 patients, most of whom had not had prior treatment (76.9%). Results demonstrated an objective response rate of 38.5% [91]. The current investigation into the use of ACT in combination with anti-PD-1 therapy is ongoing [92].

The results of a phase III multicenter, randomized trial investigating the use of TIL treatment compared to ipilimumab further support the potential role of TIL therapy for advanced melanoma, particularly in anti-PD-1 refractory patients. A total of 168 patients with unresectable stage IIIC-IV melanoma were randomized 1:1 to TIL therapy (n = 84) or ipilimumab (n = 84). The majority (86%) of these patients were refractory to anti-PD-1 treatment. Median progression-free survival was significantly longer for patients treated with TIL therapy at 7.2 months compared to 3.1 months. There was, however, a considerably higher rate of grade ≥ 3 adverse events for the patients that received TIL therapy (100% compared to 57% of patients treated with ipilimumab) [93]. Additional evidence for the potential benefit of ACT in patients with ICI-refractory melanoma can be seen in a trial utilizing lifileucel. Lifileucel is a TIL product produced utilizing a standardized protocol currently under study in advanced solid tumors, including ICI-refractory melanoma. A phase II single-arm study was conducted in 66 heavily pre-treated patients with melanoma. Patients received an average of 3.3 prior therapies, with 100% having received anti-PD-1 treatment, 80% having received anti-CTLA-4 treatment, and 23% having received a BRAF ± MEK inhibitor. After TIL harvest and production of lifileucel, the study patients underwent lymphodepletion with cyclophosphamide and fludarabine. Patients then underwent infusion of lifileucel. IL-2 was administered after lifileucel infusion. The primary outcome was objective response rate, which was 36% (CI 25 to 49%), with a sub-group analysis of patients with primary resistance to anti-PD-1 therapy demonstrating an objective response rate of 41% (CI 26 to 57%). The median duration of response was not reached at 18.7 months [94]. While ACT has not to date been FDA-approved for use in advanced melanoma, this modality provides a promising therapeutic potential for patients who are refractory to current ICI therapies.

## 9. Fecal Microbiota Transplant

The gut microbiome has been implicated as a potent modulator of innate and adaptive immunity. Approximately 70–80% of immunologically active cells are found in the mucosal immune system, most of which are present in the gastrointestinal tract [95,96]. Bacteria in the gastrointestinal tract produce microbiota metabolites and microbe-associated molecular patterns (MAMPs). MAMPs stimulate innate immunity by activating toll-like receptors, inflammasomes, C-type lectins, and RNA-sensing RID-like helicases [97,98]. The microbiome has also been shown to alter the adaptive immune response by modulating cytokine production and activating T helper type 17 cell response [99,100]. The microbiome, therefore, serves as an essential host characteristic that may play a role in the heterogeneity seen in response to anti-PD-1 therapy.

Characterization of the microbiome of patients with metastatic melanoma treated with anti-PD-1 therapy has revealed that certain bacterial species are more abundant in patients with clinical responses than those without [101,102]. These findings prompted the hypothesis that fecal microbiota transplantation (FMT) from anti-PD-1 responders to non-responders would overcome resistance to anti-PD-1 therapy. In 2021, the first clinical trial utilized FMT in advanced melanoma patients. A total of 16 patients with advanced melanoma with no response to anti-PD-1 therapy alone or in combination with anti-CTLA-4 therapy after a minimum of two cycles were enrolled. Seven FMT donors were chosen based on prior responses to anti-PD-1 treatment (4 CR, 3 PR). Samples were screened for viral, bacterial, fungal, and protozoan pathogens. Trial patients were then re-challenged with pembrolizumab along with single-donor FMT. This process is illustrated in Figure 2. Objective responses were achieved in 3 of 15 (20%) patients that completed 12 weeks of treatment. The observed adverse events were typical of pembrolizumab monotherapy [103]. The results of this trial indicate the potential of FMT for patients with anti-PD-1 refractory disease.

## 10. Conclusions and Future Directions

The advent of immune checkpoint inhibitors has significantly improved outcomes for patients with advanced, inoperable, and high-risk operable melanoma (Table 1). However, there remains a portion of patients with advanced melanoma who have progression of disease despite this treatment. Some of these patients have primary resistance to immune checkpoint blockade therapy, while others will progress after some period of stability or response to treatment. Factors that contribute to ICI resistance include tumor-induced immune suppression, upregulation of other inhibitory T cell receptors, loss of IFN-γ response elements, and loss of MHC class 1 [104,105,106,107]. A hypoxic tumor microenvironment also may lead to T cell exhaustion [108]. Investigational immunotherapies attempt to address these resistance mechanisms.

One such approach involves activating the stimulator of the interferon gene (STING) pathway. STING is a transmembrane protein that when activated initiates a signaling cascade that leads to the production of type I interferon (Figure 2). The secreted interferon generates a more robust immune response via enhancement of T cell activity and activation of NK cells. STING agonists may have a synergistic effect with ICI therapy [109,110,111,112]. Trials are currently investigating the efficacy of STING agonists alone and in combination with anti-PD-1 therapy.

ICI therapy has been shown to cause the upregulation of other inhibitory receptors. Examples include T cell immunoglobulin and mucin-domain containing-3 (TIM-3) and V-domain Ig suppressor of T cell activation (VISTA) [107,113,114]. These, and other inhibitory receptors’ activation, may suppress T cell function despite PD-1 or CTLA-4 blockade. This observation has led to the development of antagonistic monoclonal antibodies of several inhibitory receptors associated with anti-PD-1 resistance. These include T cell immunoreceptors with Ig and ITIM domains (TIGIT) and TIM-3 [115].

Chimeric antigen receptor (CAR)-T cell therapy has led to significant improvements in outcomes for patients with hematologic malignancies. Evaluation is underway to determine if CAR-T cell therapy can produce similar results in solid tumors including melanoma. However, there are additional barriers for CAR-T cell therapy in these patients, which may contribute to its limited efficacy in melanoma to date. Engineered T cells must migrate out of the vasculature and into an immunosuppressive tumor microenvironment while effectively targeting tumoral antigens. In a study using VEGFR-2 as the target antigen for CAR-T cells, 23 of 24 patients had progressive disease and one patient had stable disease while 5 of 24 patients had serious adverse events [116]. Investigation of CAR-T cell therapy utilizing other target antigens including gp-100, B7-H3, and GD2 is underway [117,118,119].

The last two decades have seen dramatic growth in the number of immunotherapy agents available for melanoma treatment, which has substantially improved outcomes. The durable responses seen in a subset of patients treated with immunotherapy serve as a signal for immunotherapy’s power in treating melanoma. Novel immunotherapeutic approaches possess the ability to overcome resistance to immune checkpoint blockade and to broaden the population of patients that experience durable benefits from immunotherapy treatment.

## Figures and Tables

**Figure 1 cancers-15-01106-f001:**
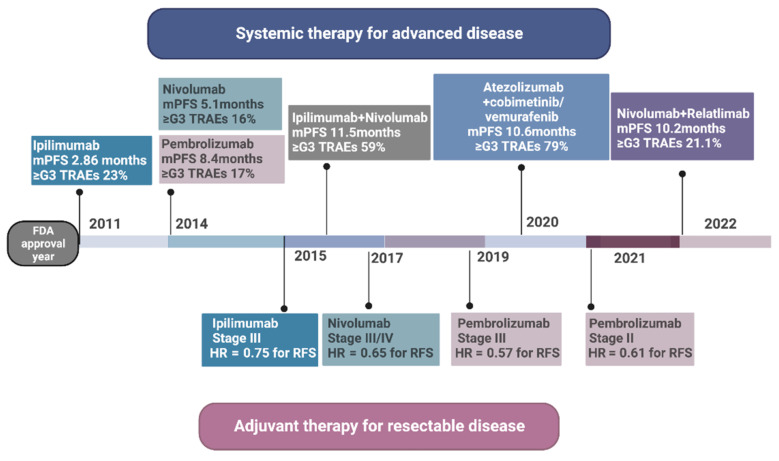
Management of melanoma with FDA-approved immunotherapy agents.

**Figure 2 cancers-15-01106-f002:**
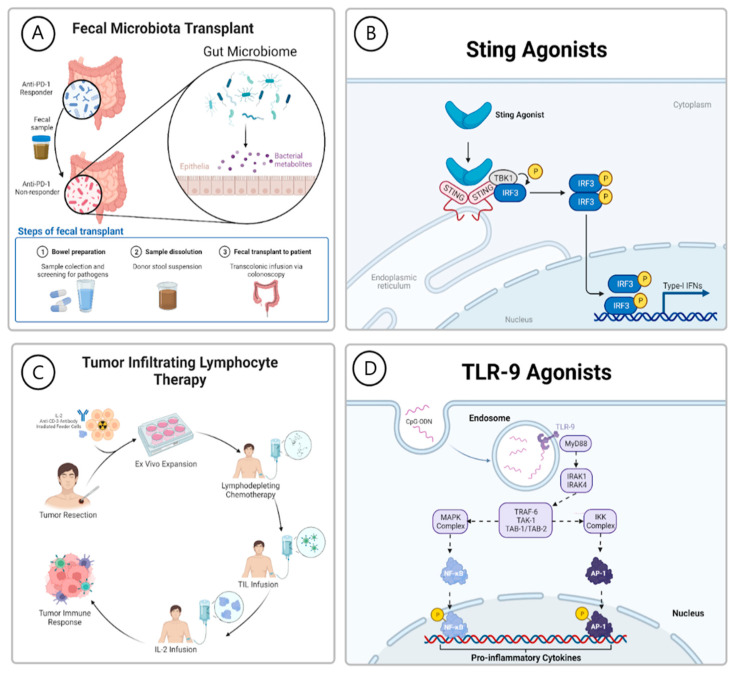
Novel immunotherapy approaches for the management of advanced melanoma. (**A**) Processing and administration of fecal microbiota transplantation. (**B**) Mechanism of STING agonists in stimulating interferon production. (**C**) Processing and administration of tumor infiltrating lymphocyte therapy. (**D**) Mechanism of Toll-like receptor 9 agonists.

**Table 1 cancers-15-01106-t001:** Clinical Trials of Immunotherapy Agents.

Medication	Study Phase	Clinical Setting	Efficacy
Ipilimumab vs. Placebo	Phase III	Adjuvant	5-year RFS: 40.8% Ipilimumab vs. 30.3% placebo HR for recurrence/death: 0.75
Nivolumab vs. Ipilimumab	Phase III	Adjuvant	1-year RFS 70.5% Nivolumab vs. 60.8% Ipilimumab HR for recurrence/death: 0.66
Pembrolizumab vs. Placebo	Phase III	Adjuvant	1-year RFS 75.4% Pembrolizumab vs. 61.0% placebo
Relatimab + Nivolumab	Phase II	Adjuvant	1-year RFS: 100%, pCR 59%
Ipilimumab + gp100 vs. ipilimumab vs. gp100	Phase III	Unresectable or Metastatic	Median OS: 10.0 mo Ipilimumab + gp100 vs. 10.1 mo Ipilimumab vs. 6.4 mo gp100
Ipilimumab + Nivolumab vs. Ipilimumab	Phase III	Unresectable or Metastatic	Median OS: 72.1 mo Ipilimumab + Nivolumab vs. 36.9 mo Nivolumab vs. 19.9 mo Ipilimumab
Relatimab + Nivolumab vs. Nivolumab	Phase II/III	Unresectable or Metastatic	Median PFS: 10.1 mo Relatimab + Nivolumab vs. 4.6 mo Nivolumab
Lifileucel (Adoptive Cell Therapy)	Phase II	Unresectable or Metastatic	Objective response rate 36% (CI 25 to 49%)
Adoptive Cell Therapy vs. Ipilimumab	Phase III	Unresectable or Metastatic	Median PFS: 7.2 mo ACT vs. 3.1 mo Ipilimumab
Pembrolizumab	Phase II	Neoadjuvant vs. Adjuvant	EFS (recurrence or progression, HR 0.59, CI 0.40–0.86)
Vidutolimod	Phase II	Neoadjuvant	Pathologic response rate: 70% (21/30 patients)
Fecal Microbiota Transplantation + Pembrolizumab	Phase II	Unresectable or Metastatic	Objective response rate: 20% (3/15 patients)
Pembrolizumab + T-VEC vs. Pembrolizumab + Placebo	Phase III	Unresectable or Metastatic	HR for progression or death 0.86, CI 0.71–1.04 HR for death 0.96, CI 0.76–1.24

Abbreviations: RFS = Recurrence-free Survival, PFS = Progression-free Survival, EFS = Event-free Survival, OS = Overall Survival, HR = Hazard Ratio, CI = Confidence Interval, pCR = Pathologic Complete Response, ORR = Objective Response Rate.

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
