# Peer review of "Immunotherapy in Melanoma: Recent Advances and Future Directions"

_cancers, 2023, doi:10.3390/cancers15041106_

Round 1
Reviewer 1 Report
Dear Authors,
You present here an interesting work focused on the use of immunotherapy in the treatment of melanoma. You present very well the stages of FDA approval for different substances of this category, as well as the stages of disease in which they are efficient.
Immunotherapy represents an important step in melanoma management, due to life extension from few months to maybe several years. Unfortunately, the association of these drugs comes usually with a growth in adverse effects.
Please detail terms like OS in line 78, RFS in line 95, HR, CI in line 101, PFS in line 133, etc.
English is fine. References are well chosen and in agreement with data presented.
I suggest that it would be interesting that you detail what are the improvements which come with the association of immunotherapy and cytostatics., for example, what are the modifications in melanoma, such as dimension, aspect, etc. of tumors.
Author Response
Thank you for taking the time to review our article. Your efforts in evaluating our work and providing constructive feedback are greatly appreciated.
The previously abbreviated terms such as OS and RFS have been changed to their full-length terms to improve clarity and readability.
We have included descriptions of the currently available therapies and highlighted investigational approaches with therapeutic potential. We specifically highlight several of the key mechanisms by which melanoma can be resistant to immunotherapy and modifications to the tumor microenvironment, which may improve response to these therapies.
Reviewer 2 Report
This review by Knight and colleagues is a well written summary of the recent clinical trials using immunotherapy to treat melanoma. Table 1 nicely summarizes the different immunotherpy trials but is missing some of the more recent trials discussed at the end, including the CMP-001 trial, Masterkey and the ACT trials. If possible, it would be good to include them also. Also, there have been some discussions lately of CAR-T therapy in melanoma. This should be discussed, although the trials have not been positive at this point.
Minor points:
1. The brackets separating the references from the text must be preceded by a space.
2. Sentences need to end with a period (that should come AFTER the reference bracket, and not before like occasionally in the text).
3. Abbreviations need to be defined when they appear for the first time.
4. The sentence in line 183 that contains „with any pathologic response“ needs revision.
5. Line 193: „were expanded upon in the CheckMate 037 TRIAL“.
6. Lines 289-291: The sentence needs improvement.
7. Line 355-356: „allowing exposure TO PAMPs.
8. Line 379: „NINE were treated at the recommended....of which SIX experienced....
9. Line 470: myeloblaStive
Author Response
Thank you for taking the time to review our article. Your comments and feedback are greatly appreciated and have been integrated into our revised draft.
Table 1 was updated to include the additional trials mentioned later in the article including CMP-001, Masterkey, ACT and FMT trials.
We have added to the discussion section in order to include CAR-T cell therapy. We highlight potential barriers to efficacy and include results from prior investigation.
Minor points:
- Brackets were separated from the text.
- Sentences were reviewed and now end with a period after the brackets rather than before as was previously the case.
- Abbreviations were removed and replaced by their respective terms
- Line 183 was revised to improve clarity
- The word trial was added
- Line 289-291 was revised to improve clarity
- Changed to add the word "to"
- Used words for single digit numbers rather than numeral (i.e. five instead of 5.)
- We utilize the term non-myeloablative to be most consistent with the referenced literature (see references 90 and 92).
We appreciate your contributions, which have improved the quality of our manuscript.